# Advances in Research on the Biological Characteristics of Weedy Rice

**DOI:** 10.3390/plants14203188

**Published:** 2025-10-17

**Authors:** Xingyi Liang, Can Zhao, Kunlun Liu, Weiling Wang, Zhongyang Huo, Xiaoling Song, Sheng Qiang

**Affiliations:** 1Jiangsu Key Laboratory of Crop Genetics and Physiology/Jiangsu Key Laboratory of Crop Cultivation and Physiology/Jiangsu Co-Innovation Center for Modern Production Technology of Grain Crops/Agricultural College, Yangzhou University, 88 Daxue South Road, Yangzhou 225009, Chinaliunan990523@gmail.com (K.L.); weilingw@163.com (W.W.);; 2Weed Research Laboratory, Nanjing Agricultural University, Nanjing 210095, China

**Keywords:** weedy rice, phenotypic diversity, stress resistance, seed shattering and dormancy, survival strategy

## Abstract

Weedy rice (*Oryza* spp.) has become one of the most harmful weeds in rice fields worldwide. It is a conspecific plant of cultivated rice (*Oryza sativa* L.) belonging to the genus *Oryza*, widely occurring in global rice production systems with a cosmopolitan distribution across major rice-growing regions. Due to its unique biological characteristics, such as strong environmental adaptability, stress resistance, seed shattering propensity, seed dormancy, and competitive dominance, weedy rice can rapidly proliferate and persist in fields, posing a severe threat to rice production systems. This review summarizes the current research progress on the biological characteristics of weedy rice and introduces the significant differences in biological characteristics between weedy and cultivated rice, such as phenotypic diversity, seed shattering, dormancy, strong competitiveness, stress resistance, and early maturity. These distinct biological traits, which significantly differ from cultivated rice, serve as essential mechanisms in the survival strategy of weedy rice. Our review will provide a theoretical reference for a deeper understanding of weedy rice and its integrated management.

## 1. Introduction

Weedy rice (*Oryza* spp.), a gramineous plant of the genus Oryza (family Poaceae), refers to rice plants that grow as weeds in paddy fields or adjacent cultivated lands, exhibiting commensal growth with cultivated rice (*Oryza sativa* L.) [1]. Given its ability to adapt to the artificial habitat of cultivated rice and persist in paddy fields, it is named “weedy rice.” Due to its close phylogenetic relationship with cultivated rice, many researchers define rice plants with weedy and feral characteristics growing in paddy fields and their surroundings as weedy rice [2,3]. Weedy rice is now recognized as a noxious weed in major rice-growing countries and regions worldwide [4,5,6], and in some temperate countries, it has become the third most significant weed after barnyard grass *(Echinochloa crusgalli*) and sprangletop (*Leptochloa chinensis*).

The occurrence of weedy rice is extensive, affecting most rice-growing regions worldwide to varying degrees (Figure 1, Table 1). It has been reported in over 50 countries and regions across the Americas, Asia, Europe, Africa, and Oceania [4,7,8,9,10]. Currently, weedy rice is distributed in states such as Louisiana, Mississippi, Texas, and Arkansas in the southern United States, where it is listed as one of the most severe weeds impacting rice production.

In Latin America, weedy rice has invaded multiple countries, including Brazil, Mexico, Bolivia, Chile, Colombia, Guyana, Venezuela, and Costa Rica [20,21,26,27].

Asia is the center of origin of rice and the continent with the largest rice-cultivated area and highest production. Weedy rice occurs across virtually all major rice-growing countries in Asia and exhibits a broader geographic range than most wild Oryza populations, with confirmed presence in Thailand, Myanmar, Laos, Vietnam, Malaysia, Nepal, Bangladesh, India, Pakistan, Sri Lanka, the Philippines, Japan, the Republic of Korea, and China [5,12,13,16,17,22,28]. Weedy rice is widely regarded as a natural hybrid between wild and cultivated rice. In Europe—including Italy, Spain, Portugal, Greece, and France—it is prevalent, and in many cases the infested area exceeds half of the national rice acreage [14,23,29]. Weedy rice is likewise a serious weed in parts of Africa’s rice-growing regions, with its distribution concentrated in West Africa and the southern Sahel. In countries where infestations are particularly severe—such as Senegal, Nigeria and Congo—a majority of rice-growing areas are affected [22,25]. The infestation of weedy rice can result in substantial yield losses, deterioration of grain quality, and increased production costs due to additional labor and herbicide use [6]. Because of its close genetic relationship with cultivated rice, controlling weedy rice is particularly difficult, and conventional weed management approaches are often ineffective. Beyond the farm level, the persistence and spread of weedy rice threaten regional and global rice supply chains, making it not only a local agronomic problem but also a food security issue of global concern [30].

Compared to cultivated rice, weedy rice exhibits richer biological characteristics, mainly manifested in phenotypic diversity, seed shattering, strong competitiveness, dormancy, and stress resistance. Additionally, weedy rice possesses the characteristic of early maturity, which enables it to cause long-term and continuous damage to rice. In order to comprehensively manage weedy rice, it is essential to clearly recognize its biological features that significantly differ from those of cultivated rice. Based on this, this study comprehensively summarizes the biological characteristics of weedy rice, explores the current research progress and potential future directions, and aims to provide scientific information for rice growers, breeders, and agricultural managers to promote sustainable agricultural development and reduce the impact of weedy rice on the rice industry.

## 2. Biological Characteristics of Weedy Rice

### 2.1. Phenotypic Diversity of Weedy Rice

Weedy rice exhibits rich phenotypic diversity, and numerous studies have been conducted on its morphology and growth characteristics (Table 1). During the vegetative growth stage and the grain-filling process, weedy rice shows significant differences in plant height and lemma color compared to cultivated rice, with rich morphological characteristics [31,32]. These distinct traits can be used to differentiate most cultivated rice varieties from weedy rice during the vegetative growth stage of rice [33]. Weedy rice has hairy leaf surfaces and extensive pigment deposition in different parts of the plant. Compared to the accompanying cultivated rice, weedy rice flowers and matures earlier, with lower thousand-grain weight [11,28,34]. Weedy rice has a canopy structure that varies from plant to plant; some populations have shown a less upright structure (sometimes sprawling), with an open canopy, than the tight and vertical canopy of rice [35].

The research team led by Wang discovered that the morphological traits of weedy rice (*Oryza sativa* f. *spontanea*) exhibit significant correlations with latitude, mean temperature, minimum temperature, precipitation, and diurnal temperature range factors. Paddy fields in China have been severely infested by weedy rice, with its morphological and biotypic differentiation demonstrating distinct geographic, climatic, and cultivated-rice-variety-dependent characteristics [36]. Yang Hongmei [37] investigated the biological characteristics of weedy rice in Leizhou, Guangdong, and found that the weedy rice had taller plants, more tillers, larger tiller angles, loose plant types, and lower seed setting rate and thousand-grain weight. It is reported that the collar and auricle of Jiangsu weedy rice were purplish-red, with brownish plant bases; the seedlings grew vigorously during the seedling stage, with more tillers and larger tiller angles than cultivated rice (over 20°), and the plants were 8–10 cm taller; the plant type was loose, and the single-plant seed setting was higher but the seed setting rate was lower [38]. Northern weedy rice exhibits rich diversity in photosynthetic and water physiological characteristics, with a wide range of photosynthetic rate variations [39]. Ningxia weedy rice has relatively small phenotypic variation and a more uniform type, with most plant characteristics similar to those of local cultivated rice [40]. Compared to cultivated rice, weedy rice generally emerges earlier, grows more rapidly, and has taller plants, more tillers and spikelets, and larger leaf areas [41,42,43,44,45]. In the Philippines, weedy rice plants are generally taller and characterized by longer, drooping leaves and panicles compared with cultivated rice. Their grains may be either awned or awnless, and possess red or white pericarps, in contrast to the awnless grains of cultivated rice [16]. In Sri Lanka, the morphological variation among weedy rice populations was greater than that within populations, with most variations originating from plant height, grain weight, panicle number, and tiller number (Ratnasekera et al. [15]). In tropical regions (e.g., India, Malaysia, Sri Lanka), weedy rice exhibits rapid seed germination due to high temperatures and humidity, compared to temperate *regions (e.g., Italy*, Spain, Japan) where dormancy periods are longer, as shown in studies from India and Europe [24].

In the United States, weedy rice has multiple flowering strategies, with early flowering in awnless, red-husked weedy rice, while black-husked, awned weedy rice flowers later than cultivated rice. The plant height of weedy rice in the United States was 47% higher than that of cultivated rice, with more tillers and spikelets [26]. Additionally, it has been found that weedy rice has an easy lodging characteristic [46]. Wang et al. [47] demonstrated that the bending stress and lignin content of indica weedy rice that is prone to lodging are lower than those of non-lodging strains. Compared with the coexisting cultivated rice, the ratio of cellulose to lignin content in lodging-prone weedy rice is lower. The genetic diversity of weedy rice (Nei’s diversity index of 0.45) is higher than that of cultivated rice (0.24) but lower than that of allelopathic rice (0.56) [48]. The genetic diversity of weedy rice determines its phenotypic diversity, and further research should be conducted on the big data correlation analysis between the two. Future research should integrate genomic, transcriptomic, phenomic, and metabolomic data to systematically analyze the genetic and phenotypic diversity of weedy rice. In Southeast Asia, a key to the recent ecological success of the weedy rice in the region is the introduction of the trait for photoperiod insensitivity from modern rice varieties into the local gene pool shared by the cultivated, wild, and weedy rice [5].

### 2.2. Seed Shattering

Seed shattering, a characteristic of natural selection, is a prerequisite for seed persistence and dispersal and spread. Seed shattering is a key characteristic that enables weedy rice to successfully adapt to farmland environments, thrive, and evade manual harvesting [49]. This trait not only enhances the survival capacity of seeds but also strengthens their regional dispersal ability. Weedy rice seeds possess a strong propensity for shattering [50], allowing them to naturally detach from the panicle axis and fall onto the soil surface, forming a soil seed bank. Seed shattering is a crucial weed characteristic that enables weedy rice seeds to escape the artificial harvesting process and enter the soil for self-propagation. In plant taxonomy, seed shattering is considered one of the main distinguishing features between weedy rice groups and cultivated rice groups [51]. The seed shattering of weedy rice facilitates its propagation and spread in paddy fields [22]. Within individual fields, both the quantity of shattered weedy rice seeds and their dispersal distance showed positive correlations with panicle density. On average, after completing harvest operations in an infested field, a combine harvester could transport over 5000 weedy rice seeds (accounting for 22.80% of total residual grains) into adjacent fields [52]. Weedy rice seeds can escape from the panicle and fall to the ground to enter the soil seed bank before harvest or due to mechanical vibration during the harvesting process. Owing to the strong dormancy and high seed vigor of weedy rice seeds, they can survive adverse environments and germinate and grow in suitable conditions during the next rice planting season, thus completing a life cycle [27,53].

Seed shedding is triggered by the continuous development of the abscission layer tissue between the floret and the pedicel [18,54]. The formation and shedding of the abscission layer in weedy rice occur earlier than in wild rice [55]. Seed shattering is related to the formation of the abscission layer and the shedding of the panicle axis in rice. The abscission layer in cultivated rice is underdeveloped and does not shed, while the abscission layer in weedy and wild rice is fully developed and sheds [19,56]. Weedy rice begins to shed seeds on average 15 days after flowering, as early as 9 days after flowering, and the shattering ability increases gradually with seed maturation, with an increasing number of shed seeds [29]. The genes qSH1 and SH4 can affect the formation of the abscission layer and are two important genes regulating seed shattering in rice [57,58]. Akasaka et al. [19] found that the seed shattering of weedy rice in Okayama, Japan, increased continuously three weeks after heading and was completely shed five weeks after heading. They also found that the sequences of the genes qSH1 and SH4 regulating abscission layer formation in weedy rice were not significantly different from those in cultivated rice, speculating that unknown genes may regulate the shattering of weedy rice in this region. It has been reported that there is one SNP site in the functional region of qSH1 between weedy and cultivated rice, and the shattering of weedy rice is unrelated to SH4 but is controlled by qSH1 [59,60,61]. Sequencing this gene (SH4) in weedy rice worldwide showed that most weedy rice strains carry the non-shattering domestication allele, suggesting the importance of other parts of the genome in the reversion to shattering; however, in the two major U.S. weedy rice ecotypes, crop × weed mapping identified seven QTLs that do not overlap with sh4 or other known loci, indicating that phenotypic parallelism in quantitative traits such as seed shattering can arise via distinct genetic mechanisms [62,63]. The genes controlling seed shattering in weedy rice are not entirely the same as those in cultivated rice. Recent studies have shown that the expression levels of *OsCPL1* and genes related to cell wall degradation, such as *OsXTH8* and *OsCel9D*, in weedy rice 10 days after flowering are significantly different from those in cultivated rice and are closely related to the shattering of weedy rice, possibly participating in the regulation of seed shattering in weedy rice [64]. The genes *OsCPL1* and *SHAT1* also participate in the regulation of rice shattering by affecting the formation of the abscission layer [65,66]. In addition, hormones are also involved in the formation of the abscission layer. Lang et al. [67] demonstrated that ABA facilitates seed shedding in weedy rice and transiently interacts with other hormones to trigger hormonal imbalance, thereby activating downstream responses in the abscission zone (see Table 2).

### 2.3. Seed Dormancy

Seed dormancy refers to the phenomenon where viable seeds fail to germinate under conditions suitable for germination. This trait enables weedy rice seeds to survive adverse conditions and wait for suitable conditions to germinate, such as preventing pre-harvest sprouting in high-temperature and rainy regions. Most weedy rice exhibits strong dormancy [68]. However, some weedy rice has weak or no dormancy, such as most Ningxia weedy rice [69,70]. The dormancy of weedy rice seeds allows them to maintain viability in the soil seed bank for a long time, which is an important guarantee for their continuous propagation. The dormancy of seeds is affected by environmental conditions, such as temperature and humidity. Rice seeds are suitable for storage in warm and dry environments [53,71]. Yu et al. [72] observed the dormancy of weedy rice in Liaoning Province under different temperature conditions and found that the temperature range required for germination is relatively wide. Morphological studies, particularly those conducted in Italy, have revealed that weedy rice exhibits variable degrees of seed dormancy. The germination rates in weedy rice are markedly low immediately after harvest, with reported rates ranging from 1.6% to 3.8% after ten days of after-ripening [73]. This low initial germination level is coupled with a tendency for increased germination over time, indicating a delayed but eventual release of dormancy that contributes to the formation of a persistent seed bank. The environmental temperature and humidity during seed development on the plant and the storage environment temperature can significantly affect the dormancy of weedy rice seeds. The dormancy of weedy rice seeds is closely related to morphological and physiological characteristics such as presence or absence of awns, lemma color, and husk color. Substances in the lemma and husk can affect seed dormancy in rice, and weedy rice with strong shattering and dark inner seed coat color has stronger dormancy [74,75]. Obviously, there is also a wide variation in the dormancy of weedy rice, which may be related to the local climate and continuous rice cropping system. It is reported that weedy rice in temperate regions (30–50° N) has a higher germination rate and weaker dormancy, while weedy rice in tropical regions (20–30° N) has the opposite, with a lower germination rate and stronger dormancy [76]. It is speculated that the dormancy of weedy rice is adapted to its growth environment.

There have also been many studies on the molecular aspects of weedy rice dormancy. A putative transcription factor, Sdr-4, located on chromosome 7 of rice, can regulate seed dormancy [77]. The dormancy in weedy rice is mainly controlled by three major genes, and qSD12 may induce seed dormancy by promoting ABA accumulation in early seed development [78]. Zhang et al. [79] studied the dormancy genes of different ecotypes of weedy rice and found 12 QTL loci related to seed dormancy, of which nine were in the same position, indicating that most dormancy genes are conserved during the differentiation of different ecotypes of weedy rice. The other three QTL loci were in different positions, indicating that certain gametophytic development genes can affect the distribution balance of dormancy gene loci. The strong dormancy of weedy rice may be of great value in improving the pre-harvest sprouting resistance of important japonica rice varieties such as “Ningjing 4” [80]. It has been reported that 92.4% of weedy rice samples exhibited high dormancy at maturity. Over time, the cumulative germination rate gradually increased and reached complete dormancy release after 284 days of imbibition, and the seeds remained viable for over 200 days after imbibition [81].

Soil burial has been reported to induce dormancy in 41–72% of initially non-dormant weedy rice seeds. The induction of seed dormancy is related to changes in hormone levels in soil-buried seeds, indicating that soil burial significantly activates the regulation of hormone production in seeds [82]. The germination rate of weedy rice seeds gradually decreases with prolonged burial time, and some weedy rice seeds can survive in the seed bank for up to five months, allowing them to re-emerge in subsequent planting seasons. Additionally, when seeds are left on the soil surface, their germination rate significantly decreases [83]. The germination rate of weedy rice is negatively correlated with the degree of dormancy and increases with increasing water availability, fluctuating between low and high temperatures. Shallow-dormant tropical japonica-derived weedy rice (TRJ) can germinate at 10 °C, while temperate japonica-derived weedy rice strains cannot [84]. Research on the dormancy genes of weedy rice and their interactions with endogenous plant hormones will be a key focus in future studies.

### 2.4. Strong Competitiveness

Strong competitiveness is a crucial characteristic of weedy rice as a noxious weed in paddy fields, causing significant damage to rice yield. The competition of weedy rice has a greater impact on secondary branches than primary branches and affects lower grains more significantly [21]. Weedy rice competition may harm cultivated rice by affecting its dry matter accumulation and grain-filling characteristics [85]. Weedy rice has strong deep-sowing tolerance, with the main driving force for seedling emergence coming from the elongation of the mesocotyl [86]. With increasing soil cover depth, the elongation of the mesocotyl can significantly improve seedling emergence rate and seedling quality. The mechanism of strong competitiveness of weedy rice may be mainly attributed to a significant growth advantage in plant height and tillering [87]. The higher photosynthetic efficiency of weedy rice seedlings ensures its competitive advantage at the early establishment stage. Dai et al. [45] found that weedy rice seedlings grow rapidly, emerging 2–3 days earlier than cultivated rice, with significantly higher plant height and biomass. The photosynthetic dynamic parameters of weedy rice seedlings are larger, with total chlorophyll content 7–33% higher than that of cultivated rice, net photosynthetic rate 8–60% higher, stomatal conductance twice that of cultivated rice, and intercellular CO_2_ concentration and transpiration rate also significantly higher. The chlorophyll fluorescence dynamic parameters of weedy rice are also significantly higher. Clearly, the photosynthetic physiological advantages of weedy rice seedlings lead to its growth advantages, thereby establishing the physiological basis for strong competitiveness during the seedling stage, ultimately causing great harm to the production of cultivated rice. Therefore, it is suggested that the control of weedy rice should start from the seedling stage, also providing theoretical basis for better control of weedy rice. Jindalouang et al. [88] found that under competitive conditions, Italian weedy rice exhibited lower and variable growth characteristics; awned weedy rice showed more and higher growth characteristics, with stronger competitiveness and phenotypic plasticity. It has been reported that weedy rice is more competitive than cultivated rice. When the ratio of weedy rice to weeds is 75:25, the number of tillers, grain yield, total dry weight, and above-ground nutrient accumulation of weedy rice are all higher than those of cultivated rice [89]. Schaedler et al. [90] assessed the relative competitiveness of cultivated and weedy rice under full and low light conditions. Plant competition was evaluated through charts and competition indices. Regardless of the light environment, weedy rice was more competitive than cultivated rice, indicating that an integrated approach is needed to control this weed. Balbinot et al. [91] found that elevated concentrations of carbon dioxide (e[CO_2_]) significantly increased the seed shedding frequency of weedy rice, thereby affecting the input of weed seeds and competitive ability, exacerbating the invasiveness and persistence of weedy rice.

### 2.5. Stress Resistance

Due to its long-term semi-wild state and natural selection, weedy rice has strong adaptability to the environment. This strong adaptability is also a prerequisite for its continued survival in agricultural ecosystems. The higher phenotypic plasticity and genetic variability of weedy rice make it more able to cope with temperature variations, intermittent water availability, soil salinity, drought conditions and increased CO_2_ concentrations than cultivated rice [3]. During this process, weedy rice has developed several unique traits, such as cold tolerance [92,93], salt-alkali tolerance [72], resistance to rice blast [94], cadmium tolerance [95], lead tolerance [96], and allelopathic ability against barnyard grass [97]. Chen [98] also found that weedy rice in Dandong has excellent cold and freeze tolerance. Yang [99] studied the cold tolerance of Chinese weedy rice populations during the germination stage and found that the germination rate of weedy rice seeds treated at 15 °C was significantly higher than that of accompanying cultivated rice, indicating that weedy rice has stronger cold tolerance than cultivated rice and may have evolved new cold-tolerant mechanisms. Xie et al. [100] found that weedy rice can quickly evolve into cold tolerance to adapt local temperaure with range extension northward. There was a coordinate occurrence of significant differentiation in cold tolerance of both weedy and cultivated rice. The cold tolerance of weedy rice was positively correlated with the latitude of the sampling point (*p* < 0.05) and negatively correlated with the annual mean temperature, extreme low temperature, and mean temperature of the transplanting month (*p* < 0.05). Furthermore, ICE1-demethylation drives the evolution in cold tolerance through mediating up-regluation of the CBF cold response transcription pathway. Ding et al. [101] found that most weedy rice had strong seedling-stage drought tolerance through evaluating 33 weedy rice samples from different regions. Baek et al. [102] revealed that buried weedy rice lasted 92.7% germination rate but cultivated rice seeds only 4.3% after those seeds buried for two years. They also found that the catalase and superoxide dismutase in weedy rice were four times that of cultivated rice, giving it a stronger ability to scavenge free radicals. Weedy rice seedlings in Liaoning had strong salt tolerance during the seedling stage and screened out weedy rice materials with better salt tolerance [103]. Suh et al. [104] tested the low-temperature germination characteristics of 1304 weedy rice samples collected worldwide and found that 18.4% of weedy rice had a germination rate higher than 50% after 15 days of low-temperature treatment at 13 °C, with 23 weedy rice samples having a germination rate exceeding 90%.

Compared with cultivated rice, weedy rice shows markedly enhanced tolerance to drought stress. In China, weedy rice and japonica weedy rice populations were more tolerant to drought, in terms of germination, than their respective indica and japonica rice cultivars; indica weedy rice was also much more tolerant than the japonica rice varieties [105]. Remarkably, weedy rice withstands drought at the reproductive stage: approximately five days of water deficit at flowering raised pollen output by about 10% in Malaysian weedy rice populations, while cultivated rice experienced an approximately 20% decline [106]. Submergence tolerance is another highly variable trait of weedy rice, whereas many cultivated varieties often exhibit poor seedling establishment under heavy rainfall, inadequate drainage, or uneven field leveling [107]. It has been reported that elevated CO_2_ concentration (700 ± 50 µmol/mol) significantly increased the growth and biomass of weedy rice, exacerbated seed shedding, and extended the survival time of seeds in the seed bank [108]. Recent studies suggested that weedy rice has novel sources of resistance to devastating rice diseases such as sheath blight (caused by *Rhizoctonia solani*) and blast (caused by *Magnaporthe oryzae*) that cause severe crop losses worldwide [109,110]. Further exploration of excellent stress-related genes in weedy rice is needed to provide rich germplasm resources for the breeding of stress-resistant cultivated rice.

In addition to tolerance to abiotic stresses, weedy rice populations have also evolved strong resistance to artificial control measures, particularly herbicides. Herbicide-resistant biotypes have been widely reported in several Asian and American rice-growing regions, most notably resistance to imidazolinone (IMI) herbicides conferred by mutations in the acetolactate synthase (ALS) gene [111,112]. Both target-site mutations and non-target-site mechanisms, such as enhanced metabolism and reduced herbicide uptake, have been implicated in conferring this resistance. The widespread occurrence of IMI-resistant weedy rice, especially in fields previously planted with IMI-tolerant hybrid rice, poses a serious challenge for chemical control and highlights the adaptive capacity of this species under intensive selection pressure [113,114]. Integrating herbicide resistance management with ecological and agronomic approaches therefore remains essential for the sustainable control of weedy rice populations.

### 2.6. Early Maturity

Early maturity is an essential characteristic for the survival and persistence of weedy rice. One of the main reasons it can escape artificial harvesting is its early maturity. Weedy rice has the characteristics of a short growth period and early maturity, which allows it to complete its reproductive growth before harvest. Combined with its strong seed shattering ability, this enables it to escape artificial harvesting, which is very advantageous for its offspring propagation. There have been many reports on the early maturity of weedy rice. During the vegetative growth stage, weedy rice is very similar to cultivated rice, but its heading and maturity stages are shorter than those of local rice cultivars. During the reproductive growth stage, it ages rapidly. Weedy rice matures early and has strong seed shattering [115,116]. The plant characteristics of weedy rice in the Jiangsu area revealed that the growth period of weedy rice was 15–20 days shorter than that of cultivated rice [117,118]. In the northeastern region, weedy rice requires a shorter time from heading to maturity, 25 days less than cultivated rice [68]. Xu et al. [119] used weedy and cultivated rice from Jiangsu, Guangdong, and Liaoning as experimental materials and found in a Nanjing common garden that weedy rice required 10–33 days less from heading to complete maturity than the corresponding cultivated rice. In India, weedy rice seeds mature within a short period and shatter immediately facilitating the buildup of weed seed bank before the farmer *gets* a chance to remove the seeds and get along with the harvest of rice crop [13].

It has been found that weedy rice in the United States matures 9–25 days earlier than conventional rice in the southern United States [29,120]. The growth period of weedy rice populations in Malaysia varied widely, with most weedy rice populations having a shorter growth period than cultivated rice, showing the characteristic of early maturity [121]. In Costa Rica, similar findings were made, with the growth period of weedy rice being 10–16 days shorter than that of early-maturing cultivated rice and 20–26 days shorter than that of late-maturing varieties [122]. Despite the above observations and reports on the rapid grain filling and early maturity of weedy rice, the intrinsic mechanisms of its early maturity, as one of the key weedy characteristics of weedy rice, have not yet been detailed. Zhao et al. [123] found that regardless of the adjustment of sowing dates, the grain filling speed of weedy rice was 7–21 days faster than that of cultivated rice in two years of study. The early maturity characteristic enables weedy rice seeds to shed before rice harvesting, becoming a core adaptive strategy for the continuation of its population. Early flowering and shortened grain filling period jointly determine the early maturity phenotype, with flowering time having stronger phenotypic plasticity. The endosperm development process of weedy rice is faster and earlier than that of cultivated rice. The active endosperm cell division period of weedy rice is 4–7 days shorter than that of cultivated rice, and the active starch accumulation period is 2–8 days shorter. The rapid development of endosperm cells and starch grains leads to a shorter grain filling period in weedy rice [124]. It has been reported that the shorter grain filling period and rapid endosperm development give weedy rice early maturity compared to cultivated rice. During grain filling, the antioxidant enzyme activity of weedy rice is lower than that of accompanying cultivated rice. Weedy rice’s ability to scavenge reactive oxygen species (ROS) is weaker than that of accompanying cultivated rice, which may be one of the reasons for the rapid cytological process of endosperm cells in weedy rice. The rapid cytological process of endosperm cells and the weaker ROS scavenging ability may be the reasons for the early maturity of weedy rice [125]. Further research is needed on the physiological mechanisms of rapid grain filling in weedy rice.

## 3. Conclusions and Perspectives

In summary, weedy rice exhibits enhanced environmental adaptability and unique survival strategies across diverse farmland ecosystems (Figure 2). Weedy rice can be interpreted through the framework of Grime’s CSR strategies [126]. Its ecological success is largely driven by a combination of competitive (C) and ruderal (R) traits. As a close relative of cultivated rice, weedy rice exhibits strong competitive ability (C) through rapid vegetative growth, high tillering capacity, and efficient resource acquisition, which enables it to outcompete crops in paddy fields. At the same time, it displays typical ruderal (R) characteristics, including early maturation, high seed shattering, seed dormancy, and prolific seed production, which enhance its adaptation to frequent agricultural disturbances such as tillage and crop rotation. Although stress-tolerant (S) traits are less dominant, weedy rice shows a certain degree of persistence under suboptimal conditions, such as nutrient limitation or water stress, indicating that S-strategies contribute to its resilience in marginal environments. Taken together, the adaptive strategy of weedy rice can be best described as a CR-dominated strategy, with supplementary S-traits that further strengthen its ecological success in diverse agroecosystems. Compared to cultivated rice, its defining biological traits include: phenotypic diversity, seed shattering, seed dormancy, strong competitiveness, stress resistance, and early maturity. These characteristics not only deepen our understanding of weedy rice biology but also advance the development of targeted control strategies.

Despite the progress made in recent years in the study of weedy rice, there are still some key issues that need further exploration. Future research should focus on the adaptive differences of weedy rice populations under different environments, such as the impact of environmental factors such as temperature, humidity, and soil type on the growth of weedy rice.

Weedy rice continues to threaten rice production worldwide due to its ecological versatility and close genetic relationship with cultivated rice. Traditional weedy rice control methods mainly rely on chemical herbicides, but long-term use can lead to increased herbicide resistance in weedy rice and have adverse effects on the environment. Because weedy rice exhibits diverse biological characteristics, effective management should be based on integrated weed management—coordinating preventative and cultural practices with prudent herbicide application and the choice of appropriate rice varieties. The implementation of integrated weed management (IWM) should be prioritized. This includes crop rotation, stale seedbed practices, optimized water and nutrient management, mechanical removal, and carefully targeted herbicide application to minimize overreliance on a single control method. Advances in genomics, gene editing, and molecular marker technologies provide new opportunities to identify key genes controlling dormancy, shattering, and stress tolerance. These tools can accelerate the development of rice cultivars with improved competitiveness and reduced susceptibility to weedy rice infestation. Future research should integrate evolutionary ecology and climate change biology to understand how weedy rice populations adapt under shifting environmental and management pressures. Applying the CSR strategy framework may help predict long-term dynamics and inform sustainable management approaches. Only through such a comprehensive approach can we reduce the threat of weedy rice and ensure the sustainability of rice production systems.

## Figures and Tables

**Figure 1 plants-14-03188-f001:**
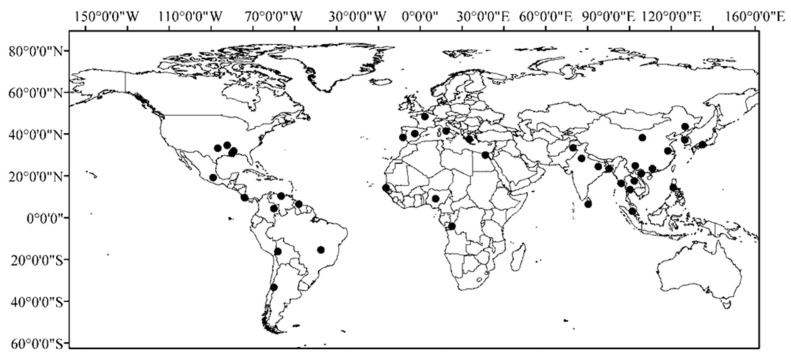
Global distributions of weedy rice.

**Figure 2 plants-14-03188-f002:**
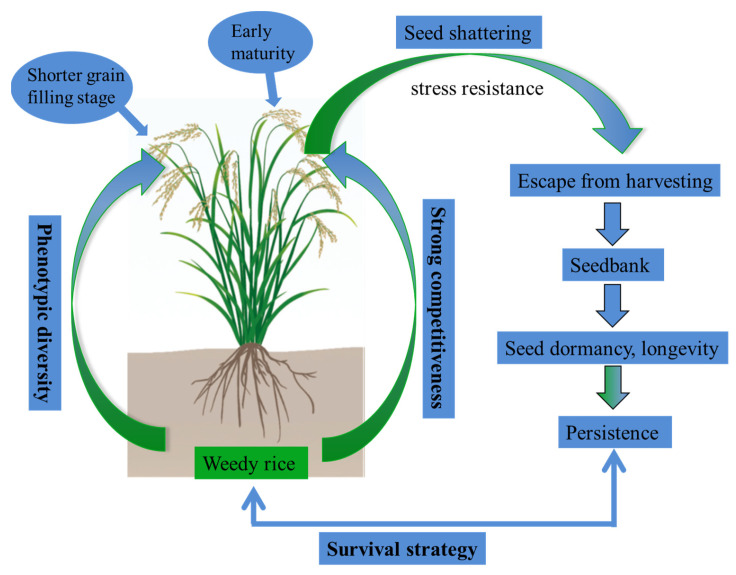
Biological characteristics and survival strategy pattern of weedy rice.

**Table 1 plants-14-03188-t001:** Global distribution, characteristics, and yield impact of weedy rice.

Region	Country/Area	Occurrence Characteristics	Biological Traits	Yield Impact
**Asia**	**China**	Widely distributed in Northeast, Yangtze River Basin, and South China; more severe in direct-seeded fields	High genetic and phenotypic diversity; early maturity; strong seed shattering	10–60% yield loss, quality deterioration [3,5,11]
	**India**	Common across major rice-growing states; associated with direct seeding practices	Rapid seed germination, dormancy variation	Up to 60% yield loss [2,3,12,13,14]
	**Sri Lanka**	Severe in traditional paddy fields; resistant to manual weeding	Strong seed shattering, seed dormancy, mimicry with cultivated rice	20–40% yield loss [15]
	**Philippines**	Widely present in irrigated and rainfed lowlands	Strong competitiveness, high tillering ability	15–40% yield loss [5,10,16]
	**Thailand**	Serious problem in direct-seeded rice systems	Early heading, high tiller production	30–60% yield loss [5,10,17]
	**Vietnam**	High infestation in Mekong Delta; worsened by direct seeding	Fast growth, strong competitiveness	20–50% yield loss [5,10]
	**Malaysia**	Severe in Peninsular Malaysia; linked to commercial seed contamination	High shattering, mimicry with cultivated rice	25–60% yield loss [3,4]
	**Myanmar**	Widespread in lowland rice areas	Early maturity, rapid seedling growth	20–40% yield loss [10]
	**Korea and Japan**	Present but less severe than SE Asia	Limited genetic variation, shattering types	10–20% yield loss [10,18,19]
**Americas**	**USA**	Severe in southern rice belt (Arkansas, Louisiana, Texas, Mississippi)	Known as *red rice*; high shattering, red pericarp	20–80% yield loss [1,6,7,8,9]
	**Brazil**	Serious in southern states (Rio Grande do Sul)	Herbicide-resistant populations, red pericarp	30–70% yield loss [20]
	**Colombia, Cuba, Venezuela**	Common in direct-seeded rice	Mimicry, dormancy, rapid seedling growth	20–50% yield loss [21]
**Europe**	**Italy**	Major issue in Po Valley rice areas (*riso crodo*)	Morphological mimicry, high shattering	30–60% yield loss, quality decline [22,23]
	**Spain**	Present in Ebro Delta and other rice regions	Mimicry, strong competitiveness	20–40% yield loss [22,24]
**Africa**	**Nigeria**	Infests irrigated and upland rice	Diverse biotypes, shattering	20–40% yield loss [25]
	**Senegal, Mali**	Serious in traditional paddy systems	Dormancy, shattering, competitiveness	15–35% yield loss [22,25]

**Table 2 plants-14-03188-t002:** Genetic and hormonal regulation of seed shattering in weedy rice.

Gene Name	Pathway/Mechanism	Function and Effect in Weedy Rice	Comparative Note (vs. Cultivated Rice)
*qSH1* [59,60,61]	Abscission Layer (AZ) Development	A major regulator promoting the formation of a complete and functional abscission layer.	Cultivated rice often has non-functional alleles of qSH1, leading to an underdeveloped AZ.
*SH4* [62,63]	Abscission Layer (AZ) Development	Transcription factor that initiates the development of the abscission layer.	A key domestication gene. Mutations in SH4 are responsible for non-shattering in most cultivated rice.
*OsCPL1* [64,65,66]	Cell Wall Degradation	Likely involved in the process of cell separation within the abscission zone. Expression is significantly higher in weedy rice.	Lower expression in cultivated rice contributes to the failure of the AZ to break down.
*OsXTH8 and OsCel9D* [64]	Cell Wall Modification and Degradation	Encode enzymes (xyloglucan endotransglucosylase/hydrolase and cellulase) that directly break down cell walls in the AZ, facilitating cell separation.	Differential expression in weedy rice is crucial for its high shattering ability.
*SHAT1* [65,66]	Abscission Layer Development	Transcription factor that affects the formation of the abscission layer.	Participates in the regulatory network controlling AZ development.
ABA [67]	Hormonal Signaling	Abscisic Acid (ABA) facilitates shedding by creating a hormonal imbalance that activates the abscission process.	Demonstrates the complex hormonal control beyond genetic factors that favors shattering in weedy rice.

## Data Availability

No new data were created or analyzed in this study. Data sharing is not applicable to this article.

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
