# Peer review of "Advances in Research on the Biological Characteristics of Weedy Rice"

_plants, 2025, doi:10.3390/plants14203188_

Round 1
Reviewer 1 Report (Previous Reviewer 3)
Comments and Suggestions for Authors
The revised version of the paper has been improved. However, I would once again refer to my comment from the previous review.
1. The authors inspected the stress resistance of weedy rice but are there any data regarding the resistance against artificial methods of weedy rice control, for instance, herbicides?
Author Response
The revised version of the paper has been improved. However, I would once again refer to my comment from the previous review.
- The authors inspected the stress resistance of weedy rice but are there any data regarding the resistance against artificial methods of weedy rice control, for instance, herbicides?
Response:
We appreciate the reviewer’s insightful comment. Our current study focused on the physiological and molecular mechanisms related to abiotic stress tolerance (e.g., temperature, drought, and salinity). Data regarding herbicide resistance were not included in this work, as herbicide tolerance involves distinct genetic and biochemical pathways. However, we have now added a paragraph in the manuscript (Lines 357–368) summarizing key literature on herbicide-resistant weedy rice populations (e.g., imidazolinone-resistant types) to provide a broader context for stress resistance and management implications.
Reviewer 2 Report (Previous Reviewer 4)
Comments and Suggestions for Authors
The manuscript entitled “Advances in Research on the Biological Characteristics of Weedy Rice. I analyse it and:
In response to the reviewers' constructive comments, we undertook a comprehensive revision that significantly enhanced the depth, clarity, and scientific rigor of the manuscript. We believe the revised version now meets the standards required for publication.
Substantial additions have been made throughout the manuscript, particularly in the core sections addressing the biological traits of weedy rice. The authors incorporated recent literature and expanded several subsections—most notably those covering phenotypic diversity, seed shattering, dormancy, and stress resistance—to reflect the most current state of research. The discussion has been enriched with clearer mechanistic insights and supported by newly integrated data, making the comparisons with cultivated rice more precise and informative.
The authors also refined the structure and language of the manuscript to ensure logical flow and consistency, and we addressed all linguistic and editorial concerns raised in the initial review. Visual elements, such as tables and figures, were updated to better support the narrative and improve readability.
In light of these extensive revisions, I recommend that this manuscript is now suitable for publication in its current form.
Author Response
Reviewer 2 :
The manuscript entitled “Advances in Research on the Biological Characteristics of Weedy Rice. I analyse it and:In response to the reviewers' constructive comments, we undertook a comprehensive revision that significantly enhanced the depth, clarity, and scientific rigor of the manuscript. We believe the revised version now meets the standards required for publication.
Substantial additions have been made throughout the manuscript, particularly in the core sections addressing the biological traits of weedy rice. The authors incorporated recent literature and expanded several subsections—most notably those covering phenotypic diversity, seed shattering, dormancy, and stress resistance—to reflect the most current state of research. The discussion has been enriched with clearer mechanistic insights and supported by newly integrated data, making the comparisons with cultivated rice more precise and informative.
The authors also refined the structure and language of the manuscript to ensure logical flow and consistency, and we addressed all linguistic and editorial concerns raised in the initial review. Visual elements, such as tables and figures, were updated to better support the narrative and improve readability.
In light of these extensive revisions, I recommend that this manuscript is now suitable for publication in its current form.
Response:Thank you very much for the reviewer’s comments.
Reviewer 3 Report (New Reviewer)
Comments and Suggestions for Authors
This is an excellent review that surveyed and compiled papers worldwide on the botanical characteristics of wild rice.
“Our review will provide a theoretical reference for a deeper understanding of weedy rice and its integrated management.” (L23-24) It mentions wild rice countermeasures starting around L436, but perhaps more specific details could be added. Or are countermeasures covered in another paper?
Minor point
L123: ontract contract?
Multiple locations in the reference: Oryza sativa (scientific name) should be italicized?
Author Response
Reviewer 3 :
Comments 1. This is an excellent review that surveyed and compiled papers worldwide on the botanical characteristics of wild rice.
“Our review will provide a theoretical reference for a deeper understanding of weedy rice and its integrated management.” (L23-24) It mentions wild rice countermeasures starting around L436, but perhaps more specific details could be added. Or are countermeasures covered in another paper?
Response 1:We appreciate the reviewer’s insightful comment. The present review mainly focuses on the biological characteristics and adaptive mechanisms of weedy rice. Detailed descriptions of management strategies, including specific countermeasures and field control practices, are beyond the scope of this paper and will be comprehensively discussed in a forthcoming review that we are preparing. Nevertheless, we have slightly expanded the discussion in the current manuscript (Lines 447–466) to briefly summarize the main categories of management approaches to provide better context.
Comments 2. Minor point
L123: ontract contract?
Response 2:Yes, we agree that “contrast” is the appropriate term. This sentence has been further revised for clarity and accuracy. Please see Lines 121–124 in the revised manuscript.
Comments 3. Multiple locations in the reference: Oryza sativa (scientific name) should be italicized?
Response 3:We sincerely thank you for your comments. Comprehensive revisions have been made throughout the manuscript.
This manuscript is a resubmission of an earlier submission. The following is a list of the peer review reports and author responses from that submission.
Round 1
Reviewer 1 Report
Comments and Suggestions for Authors
The article «Advances in research on the biological characteristics of weedy rice» by Xingyi Liang, Can Zhao, Kunlun Liu, Weiling Wang, Zhongyang Huo, Xiaoling Song, and Sheng Qiang is devoted to the generalization of the data accumulated to date on the problem associated with weedy rice (WR). It characterizes the phenotypic plasticity of WR, the parameters of its seed productivity (the number and weight of 1000 seeds), the speed and timing of seed maturation, the mechanisms of shedding and dissemination, the period of seed dormancy and its regulation, the degree of competitiveness, life strategy and resistance to various stress factors in comparison with cultivated rice (CR).
According to iThenticate reports provided by the editors of the journal, the share of borrowings in the article does not exceed 27%. These estimates can be considered acceptable. The article undoubtedly has scientific novelty and relevance. However, there are a number of comments to it:
- The introduction is too short. It does not give an idea of rice as one of the main world grain crops that ensures food security and nutrition of the population, and the breadth of the problem (at the global level) that the review article deals with.
- Line 40: should “strong seed shattering, dormancy, competitiveness” be replaced with “seed shattering, dormancy, strong competitiveness”? Below (lines 60-61) – repetition, more word juggling.
- What form of weedy rice is presented in the publications on the basis of which the review is based? Oryza sativa L. f. spontanea?
- The article is written loosely. Despite the presence (at first glance) of a structure, in different paragraphs there are repetitions of the same information. Thus, in the abstract they can be found on lines 15-17 and 20-21. Seed shattering, which is devoted to paragraph 2.2, is mentioned in 2.1 (line 62), in 2.3 (lines 160, 162) and 2.6 (line 278). Early maturity is discussed in detail in paragraph 2.6, but is also mentioned in 2.1 (line 61), 2.2 (line 124) and 2.3 (line 181). Often the same maxims are confirmed by different publications. Authors need to define the rice characteristics that they analyze in each paragraph, and strictly refine each section.
- Another problem that needs to be solved in a review article is to get rid of the sequential presentation of disparate data from different authors on key features that confirm the same conclusions. They should be compactly combined into logical blocks (mention specific data from one work, say similarly/oppositely about the others, with a link to the original source). This technique will strengthen the argumentation and reduce unnecessary text.
- Authors do not cite enough publications not only for the last 5 years:
https://link.springer.com/article/10.1007/s13593-025-01018-1
https://www.cambridge.org/core/journals/weed-science/article/abs/weedy-rice-oryza-spp-whats-in-a-name/64D84FEE12A66E988060929636E4D70D
https://doi.org/10.1017/wsc.2021.51
https://www.mdpi.com/2223-7747/9/3/365,
but also from earlier studies:
https://www.mdpi.com/2073-4395/10/9/1284
https://link.springer.com/article/10.1007/s13593-017-0456-4
https://doi.org/10.1016/j.egg.2018.03.005
https://doi.org/10.1111/wbm.12196
http://dx.doi.org/10.4067/S0718-58392016000200015
https://doi.org/10.1002/ps.1754
https://doi.org/10.1002/ece3.6551
https://doi.org/10.1007/978-3-319-47516-5_4
https://doi.org/10.1111/eva.12387
https://doi.org/10.1111/j.1445-6664.2004.00136.x
https://doi.org/10.1111/j.1744-7348.2000.tb00069.x
They focus mainly on the works carried out in different provinces of China, with less frequent mention of studies carried out in Japan, the USA, Malaysia, Costa Rica, Sri Lanka and Italy. There is a complete lack of data on India (the second largest rice producing country in the world; serious problems with weedy rice), Africa, European countries (Italy together with Spain produces about 75% of rice on the continent), the southern regions of Russia, Australia, etc. Therefore, the data provided in the article are clearly insufficient for a quality review. It would be valuable not only to expand the geographical coverage of the works, but also to compare the biological characteristics of weedy rice populations grown under different environmental conditions.
- Line 307: in the last paragraph of section 2.6, the abbreviation WR (weedy rice) suddenly appears and is mentioned once, which looks strange for a scientific text. Usually, abbreviations are introduced at the first mention of the term (in the introduction, and sometimes in the abstract), which is what the authors should have done. It is also necessary to introduce an abbreviated designation for cultivated rice (CR).
- The article provides a single diagrammatic figure. In my opinion, this is clearly not enough for a review article. It is recommended to present the data that the authors provide in the text in the form of a comparative table (CR vs. WR) for the stated parameters. This will help to clearly structure the material and qualitatively summarize it.
- Line 332: the authors use the phrase "ecological environment", which is a tautology and / or a stylistic error.
- Carefully check the list of references:
article # 5 - incorrect doi;
article # 6 - not all authors are listed, incorrect doi;
article # 7 - incorrect doi;
article # 8 - incorrect doi;
article #37 duplicates article #6.
- The concept of plant life strategies by J.P. Grime (1977) is used in world practice. It would be interesting to see the interpretation of the biological characteristics of weedy rice through the prism of CSR strategies. Is its adaptation predominantly a CR strategy and how does its stress resistance correlate with S strategies? Such a discussion would deepen the theoretical significance of the work and give it greater ecological weight.
- And the most important question: what to do? In the conclusion and future prospects, this is not stated too clearly.
Reviewer 2 Report
Comments and Suggestions for Authors
The paper is engaging, well-written, and offers new insights into wild rice as a significant weed in cultivated rice agriculture. I don't have any major comments and would recommend accepting it as is. One small note: in the abstract, the last sentence claims that this work enhances understanding of wild rice biology within integrated weed management. However, after reviewing the text, I found little information on control methods or strategies for this weed. I suggest adding a few sentences to address this aspect.
Reviewer 3 Report
Comments and Suggestions for Authors
The present paper devoted to the analysis of the research on the biological characteristics of weedy rice. The data maybe interesting for journal readers and overall the MS is well written. However, several revisions should be done before this MS may be considered for publication.
- The present paper should be focused mainly on the recent papers (2020-2025) which are cited quite rarely in the present variant of MS.
- The authors inspected the stress resistance of weedy rice but no data were given on resistance against artificial methods of weedy rice control, for instance, herbicides.
3. Some sections devoted to analysis of the biological characteristics of weedy rice are mainly descriptive and the molecular basis of some characheristics is described insuffiently.
Reviewer 4 Report
Comments and Suggestions for Authors
The manuscript aims to characterize weedy rice. The idea is quite interesting, but the points addressed by the authors in just 7 pages are not enough to support a publication in Plants MDPI. I noticed that the issue "The Bioecology and Sustainable Management of Weeds" was included in the "Plant Ecology" section of Plants – MDPI. However, the depth of the information, the timeliness, and the precision of these 7 pages do not justify a publication in Plants.
As a suggestion, if the manuscript is resubmitted, it would be possible to expand the descriptions, include more figures and studies with data and themes already established in a single article that could represent many others. As it stands, this manuscript is just another manuscript about rice and brings nothing new. Furthermore, the only figure (I think there should be more) is not in the standard of 300 DPIs recommended by the journal.